# Real-World Evaluation of Calcimimetics for the Treatment of Secondary Hyperparathyroidism in Chronic Kidney Disease, in an Italian Clinical Setting

**DOI:** 10.3390/healthcare10040709

**Published:** 2022-04-11

**Authors:** Valentina Perrone, Melania Dovizio, Chiara Veronesi, Margherita Andretta, Fausto Bartolini, Arturo Cavaliere, Fulvio Ferrante, Alessandro Lupi, Romina Pagliaro, Rita Pagnotta, Stefano Palcic, Davide Re, Loredana Ubertazzo, Adriano Vercellone, Luca Degli Esposti

**Affiliations:** 1CliCon S.r.l. Società Benefit, Health, Economics & Outcomes Research, 40121 Bologna, Italy; melania.dovizio@clicon.it (M.D.); chiara.veronesi@clicon.it (C.V.); luca.degliesposti@clicon.it (L.D.E.); 2UOC Territorial Pharmaceutical Service, Azienda ULSS 8 Berica, 36100 Vicenza, Italy; margherita.andretta@aulss8.veneto.it; 3Pharmaceutical Department, ASL Umbria 2, 05100 Terni, Italy; fausto.bartolini@uslumbria2.it; 4Pharmaceutical Department, ASL Viterbo, 01100 Viterbo, Italy; arturo.cavaliere@asl.vt.it; 5Department Diagnostic and Pharmacy, U.O.C. Pharmacy, ASL Frosinone, 03100 Frosinone, Italy; uocfarmacia@aslfrosinone.it; 6SC Cardiology, ASL VCO, 28887 Omegna, Italy; alessandro.lupi@aslvco.it; 7U.O.C. Territorial Pharmacy, ASL Roma 5, 00019 Tivoli, Italy; romina.pagliaro@aslroma5.it; 8Government Hospital Function, Local Health Unit Naples 3 South, 80059 Torre del Greco, Italy; r.pagnotta@aslnapoli3sud.it; 9Territorial Pharmaceuticals, Azienda Sanitaria Universitaria Integrata Giuliano-Isontina, 34128 Trieste, Italy; stefano.palcic@asugi.sanita.fvg.it; 10U.O.C. Local Pharmaceutical Service, Local Health Unit Teramo, 64100 Teramo, Italy; davide.re@aslteramo.it; 11U.O.C. Territorial Pharmacy, ASL Roma 4, 00053 Civitavecchia, Italy; loredana.ubertazzo@aslroma4.it; 12Department of Pharmacy, Local Health Unit Naples 3 South, 80059 Torre del Greco, Italy; a.vercellone@aslnapoli3sud.it

**Keywords:** calcimimetics, real-world data, healthcare costs

## Abstract

This Italian real-world data analysis evaluated the pharmaco-utilization of calcimimetics, cinacalcet or etelcalcetide, and the economic burden of secondary hyperparathyroidism (SHPT) in chronic kidney disease (CKD) patients. From 1 January 2010 to 30 June 2020, adult patients with: (i) ≥1 prescription of etelcalcetide or cinacalcet, (ii) ≥3 hemodialysis/week, and (iii) without parathyroidectomy, were included. Based on the drug firstly prescribed, patients were allocated into etelcalcetide- and cinacalcet-treated cohorts, and the propensity score matching (PSM) methodology was applied to abate potential cohorts’ unbalances. Overall, 1752 cinacalcet- and 527 etelcalcetide-treated patients were enrolled. In cinacalcet- and etelcalcetide-treated patients, respectively, the most frequent comorbidities were hypertension (75.3% and 74.4%), diabetes mellitus (21.0% and 21.3%), and cardiovascular disease (18.1% and 13.3%, *p* < 0.01). In covariate-balanced cohorts, the treatment adherence and persistence rates were significantly higher in the etelcalcetide-treated (80.1% and 62.7%, respectively) vs. cinacalcet-treated cohort (62.3% and 54.7%, respectively). After PSM, the total costs for the management of cinacalcet- and etelcalcetide-treated patients, respectively, averaged EUR 23,480 and EUR 22,958, with the disease-specific drug costs (EUR 2629 vs. EUR 2355, *p* < 0.05) and disease-specific hospitalization costs (EUR 1241 vs. EUR 855) in cinacalcet- and etelcalcetide-treated patients. These results showed that, in etelcalcetide-treated patients, a higher treatment adherence and persistence was found, with disease-specific costs savings, especially those related to drugs and hospitalizations.

## 1. Introduction

Chronic kidney disease (CKD) is associated with several complications [1,2,3]. Elevations in parathyroid hormone (PTH) levels, known as secondary hyperparathyroidism (SHPT), is one of the most critical complications of CKD, leading, among other various consequences, to vascular calcification, cardiovascular diseases, and death, especially in CKD patients receiving hemodialysis [4]. SHPT development is associated with a worsening imbalance in calcium and phosphorus metabolism [5], and the alterations in PTH and mineral metabolism in SHPT have been associated with increased morbidity and consequently of hospitalization in CKD patients [6,7].

Epidemiological data collected from 150 countries around the world have highlighted that the number of patients suffering from the advanced CKD was approximately 3 million at the end of 2012 [8]. In Italy, it is estimated that CKD patients on dialysis are about 48,000, with an annual incidence of 0.02% [9], and 80% of these patients on hemodialysis are estimated to be affected by SHPT [9].

Due to the consequences of SHPT occurrence, the estimated healthcare costs are more than three times higher for patients with CKD and SHPT than for those without SHPT. Therefore, treatment of SHPT may reduce the economic burden of SHPT in patients with CKD [10,11].

Current therapeutic approaches consist of phosphate intake control by diet or phosphate binders, vitamin D, and calcimimetic agents that activate the Ca^2+^-sensing receptor receptor (CaSR), the primary physiological regulator of PTH secretion, and its activation by calcium rapidly inhibits PTH secretion [12]. Calcimimetics are a newer class of agents for the treatment of SHPT in patients receiving renal replacement therapy [12,13]. To date, two calcimimetics, cinacalcet and etelcalcetide, are approved to treat SHPT in CKD patients with hemodialysis. Cinacalcet was approved in 2004, and etelcalcetide received its approval in 2016 in Europe.

Despite the availability of various treatments, the management of SHPT represents an unmet medical need; in fact, SHPT remains uncontrolled in many CKD patients, and only 15–20% of them simultaneously reach the target levels of PTH, calcium, and phosphorus [14,15,16]. One of the current therapeutic limitations is represented by the poor adherence to medications: it is estimated, in fact, that of the total number of hemodialysis patients, between 50% and 85% of them do not adhere to the treatment [17,18].

Since limited evidence is available on the use of calcimimetics among the Italian clinical practice, an observational retrospective analysis was carried out to evaluate, in a real-world setting in Italy, the characteristics of patients under calcimimetics, cinacalcet or etelcalcetide, their pharmaco-utilization, and to estimate the economic burden for the management of SHPT in CKD patients.

## 2. Materials and Methods

### 2.1. Data Source

This is a retrospective observational analysis on data extracted from the administrative databases from a pool of Italian Healthcare Departments. Data were extracted from the following databases: (i) Demographic database, which consists of all patient demographic data, such as gender, age, and death; (ii) Pharmaceuticals database, that supplies information on medicinal products reimbursed by the National Health System (NHS) as the Anatomical Therapeutic Chemical (ATC) code, number of packages, number of units per package, unit cost per package, and prescription date; (iii) Hospitalization database, which comprises all hospitalizations’ data for patients in the analysis, such as the discharge diagnosis codes classified according to the International Classification of Diseases, Ninth Revision, Clinical Modification (ICD-9-CM), Diagnosis-Related Group (DRG), and DRG-related charge (provided by the NHS); (iv) Outpatient specialist services database, which incorporates all information about visits and diagnostic tests for patients under analysis (date and type of prescription, description activity, and laboratory test or specialist visit charge); (v) Payment exemption database, which contains data of the exemption codes that allow to avoid the contribution charge for services/treatments when specific diseases are diagnosed.

An anonymous univocal numeric code was assigned to each study individual to guarantee patients’ privacy, in full conformity with the European General Data Protection Regulation (GDPR) (2016/679). The patient code in each database permitted the electronic linkage among all databases. The results were produced as aggregated summaries and never attributable to a single institution, department, doctor, individual, or individual prescribing behaviors. The analysis has been notified and approved by the local Ethics Committees of the Healthcare Departments involved in the analysis (the details are reported in the Institutional Review Board Statement below).

### 2.2. Study Design, Study Population, and Cohorts’ Definition

Among the population, in the period from 1 January 2010 to 30 June 2020 (enrollment period), adult patients (with age ≥ 18 years) with: (i) at least one prescription of etelcalcetide (ATC code H05BX04) or cinacalcet (ATC code H05BX01), (ii) performing ≥3 hemodialysis/week (identified by the main or secondary procedural codes ICD-9-CM 39.95, or specialist code 39.95), in the period comprising three months before up to three months after the index date (i.e., the date of etelcalcetide or cinacalcet first prescription), and (iii) who have not undergone parathyroidectomy (identified by main or secondary procedural code ICD-9-CM: 06.8, during six months of observation), were included. All patients who were not present in the databases during the period following the index date (i.e., relocations) were excluded from the analysis. Based on the drug firstly prescribed, patients were allocated into two cohorts: etelcalcetide-treated patients and cinacalcet-treated patients. All patients were characterized over 12 months prior to the index date and followed-up for six months after the index date.

### 2.3. Analysis of Baseline Patients’ Characteristics

For all patients included in the analysis, at the index date and during the characterization period (i.e., 12 months before the index date), respectively, demographic (in terms of age and gender) and clinical characteristics (in terms of previous treatments and comorbidities/manifestations) were evaluated. In particular, among the study populations, the presence of at least one prescription of the following medications was evaluated: angiotensin-converting enzyme (ACE) inhibitors (ATC codes C09A, C09B), sartans (ATC codes C09C, C09D), beta-blockers (ATC code C07), calcium supplementation (ATC codes A02AD01, A12AA04, A12AX, A12AA12, A12AA02), phosphorus binders (excluding calcium, ATC codes V03AE04, V03AE02, V03AE03, A02AD01), vitamin D in active form (ATC codes A11CC03, A11CC04), vitamin D in inactive forms (ATC codes A11CC05, A11AA01, A11JB, A12AX, H05BX02), erythropoietin-stimulating agents (ESA) (ATC codes B03XA01, B03XA02, B03XA03), prednisolone, and other steroids (ATC codes H02AB01, H02AB06). Among the comorbidities/clinical manifestations, the evaluation of primary or secondary discharge diagnosis or the use of specific drugs related to the following diseases was assessed: diabetes mellitus (identified by the hospitalization with primary or secondary diagnostic code ICD-9-CM 250 or by the presence of least two prescriptions of antidiabetic drugs with ATC code A10), hypertension (identified by the hospitalization with primary or secondary diagnostic codes ICD-9-CM 401–405 or by the presence of at least two prescriptions of antihypertensive drugs with ATC codes C03, C07, C08, C09), cardiovascular disease (CVD) (identified by hospitalization with primary or secondary diagnostic codes ICD-9-CM: 410–414, 428, 430–438, 440–443), fractures (identified by hospitalization with primary or secondary diagnostic codes ICD-9-CM 800–829), hypercalcemia and hypercalcemia due to ectopic secretion of PTH (identified by hospitalization with primary or secondary diagnostic codes ICD-9-CM 259.3, 275.42), total or partial parathyroidectomy (determined by hospitalization with primary or secondary diagnostic code ICD-9-CM 06.8×), or malignant tumors (identified by hospitalization with primary or secondary diagnostic codes ICD-9-CM 140–209). In addition, the comorbidity profile was assessed using the Charlson Comorbidity Index (CCI), which assigns a single score (minimum 0, maximum 6) to patients by weighting each concomitant disease identified in the 12 months before the index date [19]. The comorbidities were identified from the discharge diagnosis at the primary and secondary levels. When a diagnosis was not available, the prescriptions of specific drugs were used as a proxy to determine the specific comorbidity.

### 2.4. Pharmaco-Utilization Analysis

During the six-month follow-up, the pharmaco-utilization in etelcalcetide- or cinacalcet-treated patients was evaluated in terms of drug adherence, persistence, and dosage variation. Adherence to medication was defined by the percentage of days covered (PDC) by treatment (according to prescription supplied) over the total duration of treatment with the drug (calculated as the difference between the first and last prescription date), plus the number of days covered by the last prescription, during the six months of follow-up. The patient was considered adherent to the therapy if PDC was higher than 80% [14]. Patients were defined as persistent to treatment if they had etelcalcetide or cinacalcet prescriptions during the last two months of follow-up. The variation of dosage was evaluated during the six-month follow-up in terms of variation among dosage drug packages (for cinacalcet among 30 mg/60 mg/90 mg; for etelcalcetide among 2.5 mg/5 mg/10 mg).

### 2.5. Propensity Score Matching (PSM) Analysis

To create a balanced covariate distribution between etelcalcetide- and cinacalcet-treated cohorts, a propensity score matching (PSM) method was applied. The propensity score was estimated using a logistic regression model, considering the following confounding variables: age, sex, prevalence of treatment with cinacalcet, CCI, diabetes mellitus, hypertension, CVD, previous fractures, previous tumors, use of calcium supplements, phosphorus binders, vitamin D in active and inactive forms, use of ESA, prednisolone and other steroids, and treatment for osteoporosis. A 1:1 matching algorithm was applied to match patients in each quintile among the two cohorts, to identify two balanced and comparable cohorts. A 1:2 matching algorithm was used for the comparison between treatment-naïve patients among the two cohorts (i.e., two naïve patients treated with etelcalcetide were sampled with one patient treated with cinacalcet). After PSM, the analysis of treatment adherence, persistence, healthcare resource consumption, and costs was performed.

### 2.6. Healthcare Resource Consumption and Cost Analysis

After PSM, in alive patients treated with etelcalcetide or cinacalcet, the healthcare resource utilization during the six-month follow-up was evaluated in terms of a mean number of drug prescriptions, a mean number of hospitalizations, and a mean number of medications for outpatient specialist services. In addition, a sub-analysis considering disease-specific treatments (etelcalcetide, cinacalcet, ACE inhibitors, sartans, beta-blockers, calcium supplementation, phosphorus binders, vitamin D in active form, vitamin D in inactive forms, ESA, prednisolone and other steroids, anti-osteoporotic medications (ATC codes M05BX04, M05BA04, M05BA07, M05BA08, M05BA06, H05AA02, G03XC01, G03XC02, M05BX03), disease-specific hospitalizations (for CVD, fractures, hypercalcemia and hypercalcemia due to ectopic secretion of PTH, total or partial parathyroidectomy, kidney transplantation (identified by ICD-9-CM code 55.6), hemodialysis (by ICD-9-CM code 39.95), hypocalcemia (by ICD-9-CM code 275.41), and disease-specific outpatient specialist services [laboratory test for PTH values (procedure code 90.35.5), calcium levels (procedure code 90.11.4), phosphate levels (procedure code 90.24.3), testosterone levels (procedure code 90.41.3), creatinine levels (procedure code 90.16.3), hemoglobin levels (procedure code 90.66.2), albumin levels (procedure code 90.05.1), C reactive protein levels (procedure code 90.72.3), computerized bone mineralometry (MOC) with DEXA (dual X-ray absorptiometry) (procedure code 88.98), and hemodialysis (procedure code 39.95), was assessed.

The direct healthcare costs were evaluated over the follow-up period and were related to the following resource consumption: hospitalizations (determined by using the DRGs tariffs, overall and disease-specific), drug costs (evaluated for those drugs reimbursed by the Italian NHS, and using the INHS purchase price, overall and disease-specific), and the outpatient specialist service costs according to Regional tariffs (overall and disease-specific). Data were reported as the mean total healthcare cost per patient. Outlier costs were identified as values exceeding the mean value, three times the standard deviation (SD), and were excluded from the analysis.

### 2.7. Statistical Analysis

Continuous variables were reported as mean ± SD, and categorical variables were expressed as numbers and percentages. The results were compared between the two cohorts, and statistical significance was accepted at *p* < 0.05. All analyses were performed using Stata SE version 17.0 (StataCorp, College Station, TX, USA). According to “Opinion 05/2014 on Anonymization Techniques” drafted by the “European Commission Article 29 Working Party”, the analyses involving fewer than 3 patients were not reported, as they were potentially traceable to single individuals. Therefore, results referred to ≤3 patients were reported as NI (not issuable).

## 3. Results

Overall, from almost 8.7 million health-assisted individuals, 5651 patients were included: 5028 were under cinacalcet treatment and 623 under etelcalcetide (Figure 1). After applying inclusion end exclusion criteria, 1752 patients treated with cinacalcet (mean age 64 years, 61% male) and 527 patients under etelcalcetide (mean age 64 years, 63% male) were enrolled (Figure 1 and Table 1).

The CCI among the two cohorts averaged 1.2 and 1.0 in cinacalcet-treated and etelcalcetide-treated patients (*p* < 0.05), respectively (Table 1). The most frequent comorbidities were hypertension (in 75.3% and 74.4% of cinacalcet-treated and etelcalcetide-treated patients, respectively), diabetes mellitus (in 21% of both cohorts), and CVD, more frequent among cinacalcet-treated patients vs. the etelcalcetide-cohort (18.1% vs. 13.3%, *p* = 0.01) (Table 1). During the characterization period, in cinacalcet-treated and etelcalcetide-treated patients, the use of ACE inhibitors (26.8% and 22.6%, respectively), sartans (19.6% and 21.6%, respectively), and beta-blockers (47.0% and 51.8%, respectively) was evaluated. A higher percentage of cinacalcet-treated patients vs. the etelcalcetide cohort were prescribed with calcium supplementation (13.6% vs. 8.5%, *p* < 0.01), and Vitamin D in active form (30.7% vs. 24.9%, *p* = 0.01), while a lower percentage of cinacalcet-treated patients vs. the etelcalcetide cohort were receiving phosphorus binders (60.4% vs. 66.8%, *p* < 0.01), Vitamin D in inactive form (37.2% vs. 60.3%, *p* < 0.001), ESA (49.0% vs. 81.6%, *p* < 0.001), and prednisolone (6.7% vs. 12.7%, *p* < 0.001) (Table 1). Among cinacalcet- and etelcalcetide-treated patients, 92.1% and 52.4% were naïve to treatment, respectively (not shown).

The PSM methodology was then applied to overcome selection bias issues and compare overall cinacalcet and etelcalcetide patients (Appendix A) or patients naïve to treatments (Appendix A). After PSM, the baseline characteristics of the two cohorts were almost completely comparable, with only a slight increase of ACE inhibitor use among the cinacalcet-treated patients vs. the etelcalcetide cohort (29.9% vs. 22.2%, *p* < 0.05) (Appendix A).

Before PSM, the adherence rate in cinacalcet- and etelcalcetide-treated patients was 58.8% and 79.5% (*p* < 0.001), respectively; moreover, the persistence rate to treatment was lower in cinacalcet- vs. etelcalcetide-treated patients, 52.4% and 64.7% (*p* < 0.001), respectively (Figure 2). In covariate-balanced cohorts, 62.3% of cinacalcet-treated and 80.1% (*p* < 0.001) of etelcalcetide-treated patients were adherent to treatment, and 54.7% of cinacalcet-treated and 62.7% (*p* < 0.050) of etelcalcetide-treated patients were persistent to medication (Figure 2).

Among the etelcalcetide-treated cohort, 33.1% of patients who were not prescribed for 60 days (therefore not persistent) subsequently restarted the treatment, without abandoning therapy (not shown). During the six months of follow-up, the drug dosage averaged 35.4 ± 122.7 mg/day for cinacalcet and 1.5 ± 1.6 mg/day for etelcalcetide; in addition, 10.3% (N = 170) and 37.3% (N = 184) of cinacalcet- and etelcalcetide-treated patients, respectively, changed their drug dosage in terms of drug packages during the follow-up.

The estimation of healthcare total costs per patient was performed on matched cohorts. As reported in Figure 3A, the total costs related to all resource consumptions averaged EUR 23,480 and EUR 22,958 for the management of cinacalcet- and etelcalcetide-treated patients, respectively. Etelcalcetide-treated patients were characterized by significantly lower costs related to hospitalizations (EUR 1334) with respect to the cinacalcet cohort (EUR 1973, *p* < 0.05) (Figure 3A), while the total specialist services and drug expenditures were comparable among the two cohorts. Among the overall drug expenditures, the costs related to cinacalcet and etelcalcetide accounted for 33.5% and 42.9%, while those related to hemodialysis accounted for 90.2% and 93.8% of the overall specialist services costs in cinacalcet- and etelcalcetide-treated patients, respectively.

An additional analysis was performed by estimating healthcare direct costs related to SHPT disease (disease-specific costs). As reported in Figure 3B, the mean total disease-specific costs were comparable among the two cohorts, with the disease-specific drug costs being significantly (*p* < 0.05) lower in etelcalcetide-treated patients (EUR 2355) with respect to the cinacalcet cohort (EUR 2629) (Figure 3B).

In naïve patients, the estimation of costs for their management is reported in Figure 4. The overall costs averaged EUR 24,389 and EUR 23,121 in cinacalcet- and etelcalcetide-treated patients, respectively (Figure 4A). The hospitalization expenditures were significantly lower (*p* < 0.05) in etelcalcetide-treated patients (EUR 1526) with respect to the cinacalcet cohort (EUR 2230). The costs related to specialist services and medications were comparable among the two groups (Figure 4A).

Among the overall drug expenditure, that related to cinacalcet and etelcalcetide accounted for 29.8% and 39.9%, while the costs for hemodialysis accounted for 90.6% and 93.1% of the overall specialist services costs, in cinacalcet- and etelcalcetide-treated patients, respectively. In naïve patients, the mean total costs of disease-specific healthcare resources are reported in Figure 4B. The expenditures related to disease-specific hospitalizations were significantly lower (*p* < 0.05) in etelcalcetide-treated patients (EUR 926) with respect to the cinalcalcet cohort (EUR 1425), while the costs related to drugs and specialist services were comparable. The expenditure for cinacalcet and etelcalcetide accounted for 37.9% and 51% of the overall drug expenditure, and among the total disease-specific specialist services costs, those for hemodialysis accounted for 95.9% and 98.4% in cinacalcet- and etelcalcetide-treated patients, respectively.

## 4. Discussion

This is a real-world investigation on patients under cinacalcet and etelcalcetide treatment, focused on their pharmaco-utilization and healthcare resource consumptions, among the Italian population.

The baseline characteristic analysis of the included patients evidenced that etelcalcetide-treated patients with respect to the cinacalcet-treated cohort were characterized by a lower comorbidity index value, and also the frequency of CVD was significantly lower among the etelcalcetide-treated cohort. Due to the non-random allocation of patients among the cinacalcet- and etelcalcetide-treated cohorts, the PSM methodology was applied to the baseline covariates of the two study groups to balance them [20]. After applying PSM, covariate-balanced cohorts were defined.

The pharmaco-utilization analysis evaluated during the six-month follow-up showed a significant increase of treatment persistence, before and after PSM, respectively, in etelcalcetide-treated patients (64.7% and 62.7%) with respect to the cinacalcet-treated cohort (52.4% and 54.7%). Moreover, before and after PSM, a significantly higher adherence rate to therapy was found in etelcalcetide-treated patients (79.5% and 80.1%, respectively) vs. the cinacalcet cohort (58.8% and 62.3%). These data could be explained by several findings from the routine clinical practice which have shown that one limitation of cinacalcet use is its poor adherence to therapy, and nonadherence to cinacalcet varies from 45.6% to 71% [21,22]. In fact, in cinacalcet-treated patients, Gincherman and Park reported that proper treatment adherence in only seen in 28% and 35% of them, respectively [17,23]. The poor treatment adherence is still a significant challenge in the control of SHPT, especially in patients under hemodialysis [24]. It has been suggested that the non-compliance to cinacalcet treatment could be explained by its oral administration with the high pill burden in patients under dialysis and, in this setting, a strategy to reduce the nonadherence rate could be the simplification of the oral treatment regimen, by administering drugs intravenously during the hemodialysis session, such as the case of etelcalcetide [24].

In fact, it has been reported that intravenous administration of etecalcetide is associated with an almost 20% increase in adherence to therapy with respect to the orally administered cinacalcet [21,22,24,25,26]. Moreover, Xipell et al. have reported that 40% of patients were nonadherent to cinacalcet treatment, and these patients benefited most from the switch to etelcalcetide in the control of SHPT [27]. Moreover, socioeconomic factors, such as poor social support and lower education, have been associated with poorer medication adherence in CKD patients [28]. However, these variables were not retrievable from the administrative database.

In the present study, almost 30% of etelcalcetide-treated patients suspended the therapy for a period of at least 60 days, but subsequently restarted without abandoning treatment. This regimen could be explained by the drug indication which states that a reduction of etecalcetide dosage or temporary discontinuation of treatment may be necessary if PTH levels reach the therapeutic target range, i.e., below 100 pg/mL [29,30].

The improvement of calcimimetics’ adherence, in addition to the amelioration of clinical disease control [21], has been associated with the improvement of disease economic burden. An observational study carried out in SHPT CKD patients among the Italian population has shown that patients with better cinacalcet adherence, with respect to those with lower adherence, were characterized by a lower incidence of all-cause hospitalizations, fractures, CVD, and sepsis; moreover, despite that the management of cinacalcet-adherent patients were characterized by an increase of healthcare costs, this was almost completely offset by the reduction in costs for hospitalizations [21]. In the present analysis, after PSM, in covariate-balanced cohorts, a slight but significant decrease in hospitalization-related expenses was observed in overall and naïve patients treated with etelcalcetide vs. those under cinacalcet. In addition, we found that among the disease (SHPT)-specific costs, etecalcetide-treated patients were characterized by lower costs related to disease-specific medications. In accordance with the present results, in a recent budget impact analysis carried out on a three-year time horizon among the Italian population, it has been shown that the treatment with etelcalcetide, compared to cinacalcet, resulted in the reduction of costs for the management of clinical consequences related to the SHPT [29].

The limitations of the present analysis are related to its observational nature; thus, the results must be interpreted based on data collected from administrative databases. Our cohort of patients reflected real clinical practice by evaluating data from a subset of health-assisted individuals. In addition, there was a lack of or limited clinical information on comorbidities and other potential confounders that could have influenced the present results. Since the comorbidities analyzed herein were addressed based on any available data before inclusion (using a proxy of diagnosis), there might be incomplete capture of these variables among patients. Data related to the pharmaco-utilization analysis were derived from medical prescriptions and dispensing, and thus the reasons for nonadherence among the study populations were not recapturable in the dataset. Socioeconomic variables were not retrievable from the administrative databases.

## 5. Conclusions

In conclusion, this real-world data analysis evaluated the pharmaco-utilization and disease economic burden in patients under calcimimetic treatment in Italy. The results showed that etelcalcetide-treated patients, with respect to the cinacalcet-treated cohort, were characterized by a higher rate of treatment persistence and adherence and cost restraints, especially those related to hospitalization and disease-specific medications, which could translate into an increase in cost savings for the Italian NHS.

## Figures and Tables

**Figure 1 healthcare-10-00709-f001:**
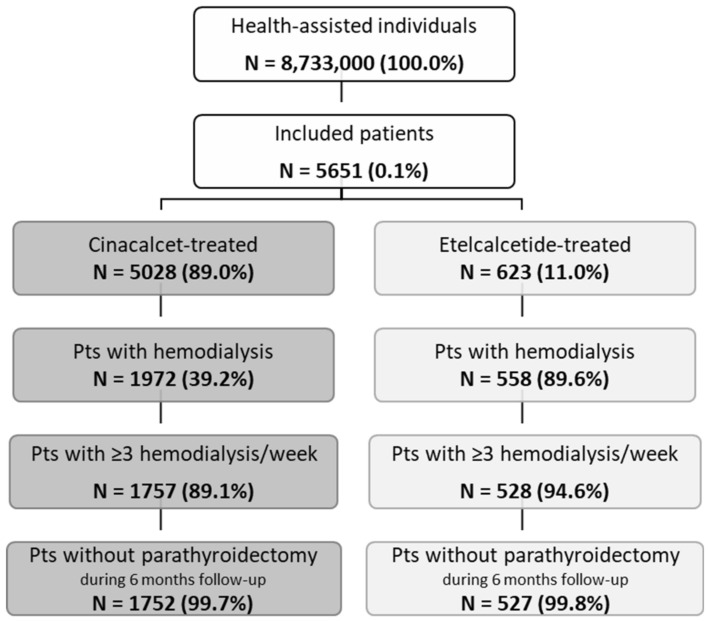
Flow chart of study population selection. Note: Pts—patients.

**Figure 2 healthcare-10-00709-f002:**
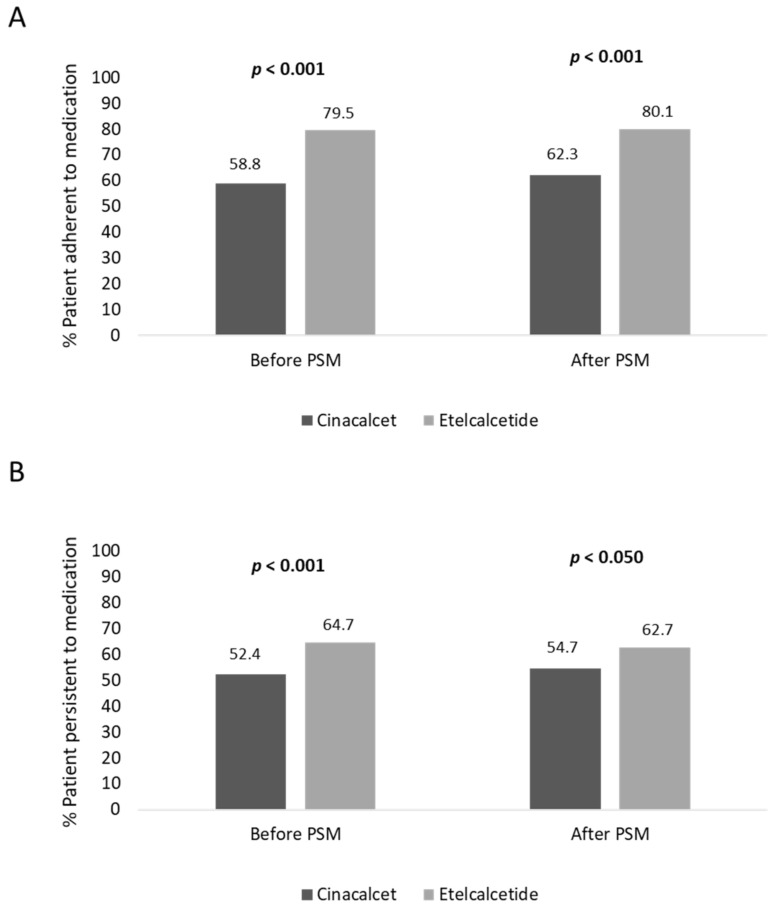
Percentage of patients adherent (**A**) and persistent (**B**) to treatment, before and after PSM analysis.

**Figure 3 healthcare-10-00709-f003:**
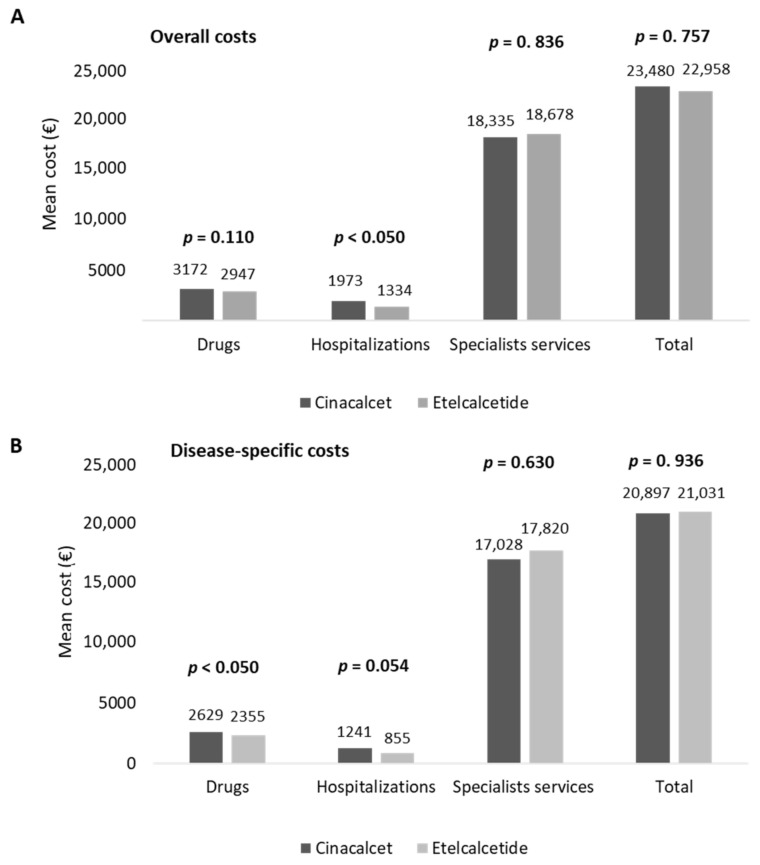
Healthcare mean total costs related to overall (**A**) and disease-specific (**B**) resource consumption in cinacalcet- and etelcalcetide-treated cohorts, post-PSM.

**Figure 4 healthcare-10-00709-f004:**
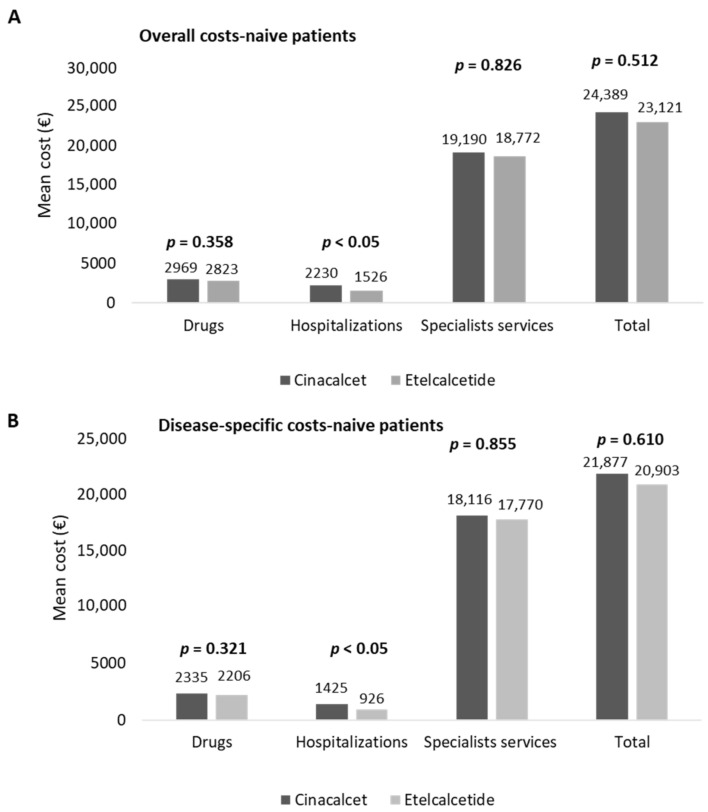
Healthcare mean total costs related to overall (**A**) and disease-specific (**B**) resource consumption in naive cinacalcet- and etelcalcetide-treated cohorts, post-PSM.

**Table 1 healthcare-10-00709-t001:** Baseline characteristics of cinacalcet- and etelcalcetide-treated cohorts.

	Cinacalcet-Treated Cohort N = 1752	Etelcalcetide-Treated Cohort N = 527	*p*-Value
Age at the index date, mean (SD)	64.0 (14.3)	64.1 (14.6)	0.877
Male, n (%)	1071 (61.1)	333 (63.2)	0.394
CCI, mean (SD)	1.2 (1.2)	1.0 (1.0)	<0.050
Comorbidities/clinical manifestations			
Diabetes mellitus, n (%)	368 (21.0)	112 (21.3)	0.903
Hypertension, n (%)	1319 (75.3)	392 (74.4)	0.675
CVD, n (%)	317 (18.1)	70 (13.3)	=0.010
Fractures, n (%)	26 (1.5)	8 (1.5)	0.955
Hypercalcemia, n (%)	NI	0 (0.0)	0.342
Parathyroidectomy, n (%)	NI	0 (0.0)	0.438
Malignant tumors, n (%)	40 (2.3)	5 (0.9)	0.054
Hypocalcemia, n (%)	NI	0 (0.0)	0.583
Treatments			
ACE inhibitors, n (%)	470 (26.8)	119 (22.6)	0.051
Sartans, n (%)	344 (19.6)	114 (21.6)	0.316
Beta Blockers, n (%)	823 (47.0)	273 (51.8)	0.052
Calcium supplementation, n (%)	238 (13.6)	45 (8.5)	<0.010
Phosphorus binders, n (%)	1059 (60.4)	352 (66.8)	<0.010
Vitamin D in active form, n (%)	538 (30.7)	131 (24.9)	=0.010
Vitamin D in inactive form, n (%)	651 (37.2)	318 (60.3)	<0.001
ESA, n (%)	859 (49.0)	430 (81.6)	<0.001
Prednisolone and other steroids, n (%)	118 (6.7)	67 (12.7)	<0.001
Anti-osteoporotic medications, n (%)	31 (1.8)	7 (1.3)	0.488

Note: CCI, Charlson Comorbidity Index; CVD, cardiovascular disease; ESA, erythropoiesis-stimulating agents; SD, Standard Deviation; NI, not issuable.

## Data Availability

All data used for the current study are available upon reasonable request to CliCon S.r.l. Società Benefit, which is the body entitled to data treatment and analysis by Local Health Units.

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
