# Peer review of "Real-World Evaluation of Calcimimetics for the Treatment of Secondary Hyperparathyroidism in Chronic Kidney Disease, in an Italian Clinical Setting"

_healthcare, 2022, doi:10.3390/healthcare10040709_

Round 1

Reviewer 1 Report

Perrone et al. present an interesting paper about the real-world evaluation of calcimimetics for the treatment of renal bone disease. This is well-written article and will add to the controversies of this difficult area of treatment.

Author Response

RE: we wish to thank the Reviewer for the positive comment.

Reviewer 2 Report

The figures and charts need to be redrawn to be more readable.

The discussion part is very confused and it should be discussed in depth

It is better to increase the literature review (introduction) with more cited papers

Author Response

The figures and charts need to be redrawn to be more readable.

RE: The figure and chart have been redrawn for more readability.

The discussion part is very confused and it should be discussed in depth

RE: As suggested by the Reviewer, the discussion paragraph has been revised.

It is better to increase the literature review (introduction) with more cited papers

RE: As suggested by the Reviewer, the Introduction paragraph has been revised.

Reviewer 3 Report

Nice representation of pharmaco-utilization of calcimimetics

Add  details of the assessment of treatment/medication adherence, Results related to treatment/medication adherence not represented properly given the parameter is discussed in the Discussion and Conclusion Sections. 

Author Response

Nice representation of pharmaco-utilization of calcimimetics

RE: we wish to thank the Reviewer for the positive comment.

Add  details of the assessment of treatment/medication adherence, Results related to treatment/medication adherence not represented properly given the parameter is discussed in the Discussion and Conclusion Sections. 

RE: As suggested by the Reviewer, in the Results paragraph the Figure 2 reporting the pharmaco-utilization data has been added.